# Effects of Cycling Dual-Task on Cognitive and Physical Function in Parkinson’s Disease: A Randomized Double-Blind Pilot Study

**DOI:** 10.3390/ijerph19137847

**Published:** 2022-06-26

**Authors:** Karina Pitombeira Pereira-Pedro, Iris Machado de Oliveira, Irimia Mollinedo-Cardalda, José M. Cancela-Carral

**Affiliations:** 1HealthyFit Research Group, Department of Special Didactics, Faculty of Education and Sports Sciences, University of Vigo, 36004 Pontevedra, Spain; chemacc@uvigo.es; 2HealthyFit Research Group, Department of Functional Biology and Health Sciences, Faculty of Physiotherapy, University of Vigo, 36005 Pontevedra, Spain; irismacoli@uvigo.es (I.M.d.O.); imollinedo@uvigo.es (I.M.-C.)

**Keywords:** Parkinson’s disease, cycling, dual task, cognitive-motor interference

## Abstract

(1) Background: Those with Parkinson’s disease (PD) may present difficulties in performing dual tasks (DT). The use of DT during training can improve different abilities. Therefore, the objective of this study is to verify the influence of a cycling exercise program combined with a cognitive task on cognitive and physical PD aspects; (2) Methods: A double-blind, randomized pilot study was undertaken. Participants performed a DT intervention composed of cycling and a cognitive task. The cycling parameters, MDS-UPDRS, PDQ-39, TUG Test, 30 s Chair Sit to Stand test and Stroop were used to measure outcomes; (3) Results: DT generated impairment in performing the cycling task, with significant differences in cycling parameters, active and passive distance (m), total work (W) and active speed (rpm). At the cognitive level, there was a trend of improvement in the group that performed the training with DT, which improved by 211%; (4) Conclusions: Combining cycling with a cognitive task caused impairment in the performance of the physical task and an improvement at the cognitive level. Therefore, combining cycling with a cognitive task in a presumably safer environment for patients with PD can be a good way to train these patients for the dual-task challenges with practical applications.

## 1. Introduction

Parkinson’s disease (PD) is one of the fastest growing neurological diseases [1], affecting around 10 million people worldwide [2]. PD is a complex disease affecting posture, gait, movement patterns and speech having a far-reaching impact on the activities of daily life [3]. People with PD may experience a wide range of non-motor symptoms, such as gastrointestinal disorders, autonomic dysfunctions, sleep problems, sensory manifestations, neuropsychiatric and cognitive symptoms, contributing negatively to severe disability and reduced life expectancy [4].

One of these non-motor symptoms is impaired executive function, which refers to a set of higher-order cognitive processes that enable goal-directed behavior and adjustments to novel situations by exerting top-down influence on lower-level cognitive processes [5]. When executive functions are impaired, behavior becomes uncoordinated and disinhibited, rendering the individual inflexible and susceptible to distraction [6].

Due to this, the population with PD may experience difficulties in performing dual tasks (DT), for example, in simultaneously combining physical and cognitive tasks [7]. The performance of DT depends on the ability to perform motor tasks automatically and the cognitive ability to combine different types of tasks [8]. 

In PD patients, the dopamine deficiency in the basal ganglia loops causes the integration of cognitive and motor information to be impaired. Therefore, the limited processing of motor-cognitive information results in inadequate responses to stimuli during movement in complex everyday situations and DT situations [9]. 

The ability to perform DT is important for performing the activities of daily life, as impaired DT ability leads to an increased risk of falls and gait freezing [10] and increased functional dependence [11]. In contrast, DT skills respond well to different behavioral training methods that reduce falls and gait freezing [12,13], giving hope that DT skills and associated disadvantages can be treatable. 

Recent review studies have explored the effects of DT in PD patients and proposed that the use of DT during training can improve gait performance, balance ability and other motor skills in PD patients [14,15]. However, studies that analyze the effect of DT on the cognitive dysfunction associated with the physical aspects of PD are not easily found. Therefore, the objective of this study was to verify the influence of a cycling exercise program combined with the performance of a cognitive task, on the cognitive and physical aspects in PD patients. To carry out this investigation, we carried out a double-blind, randomized pilot study, with parallel groups, with a 1:1 allocation rate and an exploratory framework.

## 2. Materials and Methods

This is a double-blind, randomized, pilot study that followed the guidelines of Standard Protocol Items: Recommendations for Interventional Trials (SPIRIT). It was carried out at the Laboratory of HealthyFit Research Group at the Faculty of Education and Sports Science of the University of Vigo. This study was approved by the Research Ethics Committee of the “Consellería de Sanidade da Xunta de Galicia, under protocol number 2020/057, and registered as a clinical trial at www.ClinicalTrials.gov, accessed on 4 May 2022 (ID: NCT05184608). The ethical standards outlined by the Declaration of Helsinki were followed, and all the participants were informed and gave their written consent prior to participating in the study. 

### 2.1. Participants

Patients of both sexes diagnosed with idiopathic PD who are members of the “Association of Parkinson de la Provincia de Pontevedra” and who met the selection criteria were recruited: confirmed as diagnosed with idiopathic PD of a stage within 2–3 H&Y [16]; no medical history of dementia, neurological deficits (e.g., sequelae of stroke or severe spinal cord injury) or any other preexisting condition that may limit limb movement (such as patients with a history of major surgery); no history of medical or surgical intervention that could interfere with motor function; no medical contraindications for performing therapeutic exercises; and who had results in the Mini Mental State Examination [17,18] of greater than 25 (there is no cognitive impairment—doubts of possible cognitive impairment). 

### 2.2. Randomization

A random number was assigned to each of the participants and only the principal investigator had access to the coding system. The participants were randomly allocated (1:1) to an experimental group (EG) or a control group (CG). Randomization was computer generated in random blocks using the randomization.com system. The process was carried out by a volunteer not linked to the research, preserving the anonymity of the allocation, randomly separating the individuals into EG and CG. The volunteer also prepared sealed envelopes, using codes to represent the groups. 

Only the researchers responsible for conducting the training were aware of the meaning of each code and the allocation of participants. They opened the envelope corresponding to each patient’s number before the patients started training. 

The researchers responsible for the initial assessment and reassessment were not informed of the allocation during data collection and statistical analysis. Data analysis was performed by a person who did not participate in the intervention, who only had access to the coding and was not informed as to which group each code corresponded to. 

### 2.3. Instruments

The movement therapy device MOTOmed Viva 2 Parkinson System created by the company RECK was used to carry out the intervention and to record the parameters of the cycling performed in each session. The device used was an intelligent exercise bike, with motorized mechanisms on which non-ambulatory or disabled patients can move their legs repeatedly, assisted or forced, while sitting in their wheelchair or in a regular chair [19]. The device records exercise data on a memory card, such as Active Distance (m), Passive Distance (m), Total Work (W), Active Speed (rpm), Passive Speed (rpm), Initial Left Symmetry (%), Final Left Symmetry (%), and Initial Heart Rate (HR) and Final HR (bpm). Regarding the initial and final symmetry, the values referring to the left side are presented, because one side must be taken as a reference, the values of the right side being complementary to those indicated for the left side, bearing in mind that the ideal result is 50% for each side. 

Primary outcomes are described first. To assess quality of life, the Spanish version of the Parkinson’s disease quality of life questionnaire (PDQ-39) was used [20,21]. This questionnaire consists of 39 items with five response alternatives (0 = never, 1 = occasionally, 2 = sometimes, 3 = often, 4 = always/unable), comprising of eight dimensions: mobility and the activities of daily living, as well as the subjects´ emotional well-being, stigma, social support, cognitive impairment, communication and bodily discomfort [22]. Each domain is expressed as a percentage of its maximum score, so that the higher the score, the greater the deterioration in quality of life. 

PD symptomatology was assessed using the adapted Spanish version of the Movement Disorder Society’s Unified Parkinson’s Disease Rating Scale (MDS-UPDRS) [23]. The MDS-UPDRS total score range is 0 to 200, with higher values indicating a greater impact of PD symptoms. It is noteworthy that this scale is composed of 4 parts: (I) non-motor aspects of daily life; (II) motor aspects of daily life; (III) motor aspects of the disease; and (IV) dyskinesia. In this study, we analyzed the total score and the motor aspects of the disease. 

Dynamic balance was assessed using the Timed Up and Go (TUG) test, which consists of getting up from a chair with arms, walking 3 m towards a cone, turning around the cone and returning to the starting position and sitting down again in the shortest possible time. This test is primarily used as a measure of mobility but is also useful as a measure of bradykinesia and the risk of falling while walking [24]. The test was performed three times: (1) walking normally, (2) walking saying animal names and (3) walking reciting a mental calculation (30−3 = 27, 24, 21…) [25,26]. Under the three conditions, the TUG test was performed while wireless inertial sensing devices with Wiva scientific sensors were installed between the L4 and L5 vertebrae (LetSense Group, Castel Maggiore, Italy). Wiva is a small inertial sensor (35 × 37 × 15 mm) that uses a wireless Bluetooth connection. Inside there are 9 inertial measurement units, an altimeter and a Global Positioning System, which allow the recording of information about the angular velocities reached during the TUG test. Wiva provides data on the split times obtained in the main phases and the angular velocities reached during TUG and the total time required to complete the task. 

Lower limb strength was assessed by the 30 Second Chair Stand Test, belonging to the Senior Fitness test battery created by Rikli and Jones [27]. This test consists of getting up from and sitting down in a chair as many times as possible in 30 s. The arms should be crossed over the chest and the person should sit fully, transferring their weight to the chair and then fully stand up to hip extension. 

Finally, as a secondary result, the Stroop Color-Word Test was used to measure mental speed and response inhibition. In the Stroop Color-Word Test, subjects are asked to name the ink color on which a word stimulus is printed, and the level of conflict is manipulated by varying the task’s irrelevant property of the stimuli (in this case, the meaning of the words), from conflicting or “incongruous” properties (e.g., the word “Red” printed in green ink) to neutral or “congruent” non-conflicting properties (e.g., the word “Red” printed in red ink). The test was performed with three sheets, the patient having 45 s to perform the task printed on each sheet. On the first sheet, three words were written arbitrarily in black ink: red, blue and green, and the patient simply had to read them, the number of correct answers made in this way was identified as words (W). The second card was made up of colored symbols of red, blue and green; the patient had to identify the colors, with the number of correct answers in this form being identified as colors (C). Based on the results obtained in these first two sheets, a mathematical formula was used to calculate the Colors-Words Planned, which would be considered the number of correct answers that the patient should get right on the third sheet (P × C/P + C = Colors-Words Planned). Finally, on the last sheet, the intention was that the patient was able to read each word without making mistakes, while taking into account that the written terms—the names of colors—were written in colors that did not correspond to their original meaning. The final result of the test is the interference calculated by the difference between Colors-Words Planned and the result obtained in the last sheet [28]. 

The assessment of disease symptoms, quality of life, cognitive function, dynamic balance, and strength was performed twice: once before starting the intervention (pre-intervention/pre-test), and again after the last session (post-intervention/post-test). Assessments were performed by a physical therapist who was not involved in the intervention. 

### 2.4. Intervention

A cycling program was undertaken, using the MOTOmed Viva 2 Parkinson system device. Each intervention session lasted 20 min, and comprised of 5 min of passive cycling as a warm-up (2 min at 40 rpm, 2 min at 70 rpm and 1 min at 90 rpm), 13 min of active-assisted cycling (90 rpm) and 2 min of passive cycling to cool down (40 rpm). Sessions were held twice a week, and always with at least 2 days of rest between sessions. The program lasted 7 weeks and a total of 14 sessions were carried out. 

Both experimental and control groups performed 20-min exercise programs using the MOTOmed Viva 2 Parkinson system. The difference between them was that the EG combined cycling with a cognitive task, while the CG performed only the physical cycling exercise. 

The cognitive task was a PowerPoint presentation structured coherently alongside the phases of the cycling program. The passive phases (5 min of warm-up and 2 min of passive cooling) were performed while the patients observed images of a close family environment. For the active phase (13 min of assisted active cycling) the PowerPoint presentation was structured with tasks related to: (1)Orientation (consisting of 3 tasks, for example: temporal orientation, spatial orientation)(2)Memory (consisting of 2 tasks, for example: memorizing objects/animals)(3)Calculus (comprising of 4 tasks, for example: geometry, addition, subtraction and multiplication)(4)Language (comprising of 4 tasks, for example: spelling backwards, ordering sentences)(5)Similarities (consisting of 2 tasks, for example: grouping foods of the same color).

Participants had to pay attention to the tasks and think about the answers. In each session, different task contents were presented, totaling 14 different cognitive tasks. 

Failure to comply with at least 80% attendance in the intervention resulted in the exclusion of participants from the final analysis of the results. In addition, to improve adherence to the intervention, alternative days were established for the sessions, whilst always maintaining a period of two days of rest between sessions. 

### 2.5. Data Analysis

To retain data from all the randomly allocated participants, we performed an intention-to-treat analysis. One patient was lost from the CG due to a fall that the patient suffered at home, which made it impossible to continue with the intervention protocol. All variables were expressed as means ± standard deviation (SD). Continuous data were tested for normality using the Shapiro–Wilk test and found to be normally distributed. To verify that the control group and the experimental group were statistical twins, Student’s *t*-test for independent samples was applied for age and Hoehn and Yahr stage variables. A multivariate analysis of variance (MANOVA, 2 × 2) adjusted for age and Hoehn and Yahr stage was used with the intention of determining the effect of the intervention program on the parameters of the physical exercise performed, quality of life (PDQ-39), symptomatology variables (MDS-UPDRS), dynamic balance (TUG test), strength (30 Second Chair Stand Test), heart rate and cognitive function (Stroop). A descriptive analysis of the results obtained for each group was carried out on the variables characterized by the intervention program (number of sessions, work, speed and distance). Statistical significance was set at *p* = 0.05 (two-tailed). All analyses were made with SPSS for Windows 25.0 software (SPSS Inc., Armonk, NY, USA).

## 3. Results

Fifteen individuals with PD participated in this study, three of them were women, with H&Y stage 2–3. Participants were randomized into two experimental groups (Table 1).

The data presented in Table 2 show significant differences in four of the nine intervention parameters analyzed, showing that the DT impaired the performance of individuals in physical exercise. 

Table 3 presents the comparative analysis of primary outcomes for the two groups, both before and after the intervention. The evaluation of the quality of life was carried out using the PDQ-39 questionnaire. There were no significant differences in the numbers, but the data show a deterioration in the EG (−3.92%), while in the CG there was an improvement (9.05%). The data show that there were significant differences in MDS-UPDRS between the groups in relation to the motor aspects of the disease, but not in the total score of the tool, although a trend of improvement is observed in both groups.

TUG was carried out on three occasions: (1) walking normally, (2) walking whilst saying animal names and (3) walking whilst reciting a mental calculation. The differences between the groups were recorded for the three conditions. Despite not being significant, CG improved the time in the three conditions, while EG deteriorated in conditions 2 and 3 (where there was a physical and cognitive task) and improved in condition 1, the physical only TUG. The strength of the lower limbs was evaluated by means of the 30 Second Chair Stand Test, where both groups presented an improving trend post-intervention.

The Stroop Color-Word Test was used to measure mental speed and response inhibition, the results show a very important change in the EG (211.65%), while the CG deteriorated (−33.87%) post-intervention (Table 4).

## 4. Discussion

The objective of this study was to verify the influence of a cycling exercise program, combined with the performance of a cognitive task, on cognitive and physical aspects in PD patients. The literature from studies that combine cycling with DT in patients with parkinsonism is scarce. Additionally, DT rehabilitation exercises or multimodal activities are not always recommended for these patients, as the presence of cognitive tasks can lead to the freezing of gait, loss of balance and increased falls [29]. We believed that it would be interesting to combine the safety of static cycling—in relation to falls—in order to verify its combination with DT. 

Regarding the performance of the cycling task during the intervention, the EG presented worse results in active and passive distance, total work and active speed when compared to the CG. Studies on patients with PD indicate that limited processing of motor-cognitive information results in inadequate responses to stimuli during movement in complex everyday situations and DT situations [9]. In our study, the EG patients were not able to perform physical activities with the same effectiveness as the CG, possibly because they were not able to process the performance of a physical task, cycling, when combined with a cognitive task, DT. 

Regarding the development of the cycling task, the concomitance of the cognitive task generated a difficult element in its performance, although without generating, for example, changes in symmetry between hemi bodies or Heart Rate. In the study by Chang et al. [30], gait parameters were evaluated while Parkinson’s patients performed different cognitive tasks associated with walking; one of them related to calculations, another with spatial orientation and finally one was carried out using the STROOP test. In that study, the total cycling time was 30 min for five–eight sessions, 35 min for nine–twelve sessions, and 40 min for thirteen–sixteen sessions. This means that those patients had participated in an intervention with a cycling protocol; however, they were assessed using a walking dual task and not a cycling dual task. Unlike in our pilot study, where we observed a deterioration of cycling performance, the aforementioned authors [30] observed improvements in different gait parameters while the cognitive task was being performed. It should be noted that, in our study, the different measurement tools were used at the beginning of the study and once again, seven weeks after the intervention had ended, except for the cycling performance parameters, which were collected at the same time as the cycling + cognitive task was performed. In a subsequent study by the same authors [31], the patients completed the cognitive tasks on a stationary bicycle. However, as it was a cross-sectional study, the effect of an intervention based on the dual task cannot be assumed, as the results are only recorded for the acute effect of the dual task on cycling and walking performance. On this occasion, the authors observed a deterioration of cycling performance and a deterioration also of walking during the cognitive task, this deterioration being even more marked during walking. 

It should also be noted that in our study the results of the STROOP test, when the pre- and post-intervention measurements were compared, showed a deterioration in the CG and an improvement in the EG. These results seem to indicate a positive effect on the cognitive aspect of the patients who undertook the dual task, although we did not observe any improvement, or deterioration, in the physical tests which were carried out (TUG and Sit to Stand Test). It is important to note that as part of the executive functions, the inhibition response and mental speed integrate all the neural circuits responsible for intentional and voluntary control that avoid the interference of non-relevant information with ongoing responses or response patterns [32] so it can be considered to be of paramount importance in the DT performance in patients with PD.

The mechanisms responsible for the interference between walking and the cognitive task in Parkinson’s patients are still unclear [33], and even less so are the mechanisms related to the cycling task carried out concurrently with a cognitive task. In recent years, very few studies have evaluated the effects of dual tasks associated with cycling, and the few that exist assess the acute effect of dual tasks with cycling [34,35,36] but not its possible long-term effect. As already mentioned in the introduction section, when two tasks are carried out at the same time, a competition for limited control resources seems to occur, which might generate a commitment towards the performance of one of them. In addition, there seem to be differences in dual task performance based on individual differences and other factors such as motivation, fatigue or arousal [33].

Al-Yahya’s et al. meta-analysis [37] showed that in healthy individuals there is a strong association between age and decreased gait speed during dual tasks and between cognitive level and decreased gait speed during dual tasks. In the study by Johansson et al. [38], it was observed that the performance of the dual task generated a deterioration in a significant percentage of the variables associated with gait, while without observing variations in symmetry, as we also observed in our study. These authors also observed a longer reaction time during the dual task and also with significant individual variability, which reached a deterioration of almost 40% in some individuals [38]. De Oliveira Silva et al. [39] suggest that poor DT and spatial navigation abilities are present in partial functional dependence in instrumental activities of daily living, and these tasks should be considered as a functionality screening tool in patients with dementia.

Our study, being a pilot study, consisted only of a small sample and unfortunately, in some aspects, it presented a lot of variability. The results observed, however, will help us to begin to establish a line of action in relation to research on the long-term response to dual tasks including cycling and on what aspects a dual task intervention may or may not benefit patients with Parkinson´s.

## 5. Conclusions

With this pilot study, we can conclude that the DT, as predicted, was detrimental to the performance of a physical cycling task and that the idea of combining cycling with a cognitive task in a supposedly safer environment for PD patients has benefits. At a cognitive level, the trend for improvement was in the group that performed the cognitive training. 

We can also indicate the need to deepen the research and suggest carrying out studies with a greater number of participants, as it may be productive to train and improve the ability of performing DT in patients with Parkinson´s disease, combined with cycling as the physical component.

## Figures and Tables

**Table 1 ijerph-19-07847-t001:** Baseline demographics and clinical characteristics of the participants.

	Parkinson’s Disease Participants (*n* = 15)	Experimental Group (*n* = 8)	Control Group (*n* = 7)	T Student
Age (years)	68.13 ± 7.82	70.50 ± 9.24	65.43 ± 5.22	t = 1.280; *p* = 0.223
Gender (male/female)	12/3	6/2	6/1	-
Hoehn and Yahr stage	2.33 ± 0.62	2.38 ± 0.74	2.29 ± 0.49	t = 0.270; *p* = 0.791

**Table 2 ijerph-19-07847-t002:** Cycling program physical parameters.

	Groups		T Student
	Experimental Group (*n* = 8) Mean ± sd	Control Group (*n* = 6) Mean ± sd	
**Active Distance (m)**	47,100 ± 31,400	76,910 ± 18,530	t = 2.060; *p* = 0.043 *
**Passive Distance (m)**	57,590 ± 23,930	34,800 ± 5,670	t = 2.264; *p* = 0.032 *
**Total Work (W)**	266.33 ± 176.13	418.82 ± 132.22	t = 1.997; *p* = 0.047 *
**Active Speed (rpm)**	78.18 ± 41.85	119.26 ± 28.29	t = 2.066; *p* = 0.049 *
**Passive Speed (rpm)**	83.64 ± 22.02	80.03 ± 46.23	t = 1.066; *p* = 0.205
**Initial Left Symmetry (%)**	45.07 ± 25.98	64.30 ± 15.09	t = 1.610; *p* = 0.133
**Final Left Symmetry (%)**	48.91 ± 22.36	52.15 ± 18.17	t = 0.554; *p* = 0.549
**Initial Heart Rate (bpm)**	92.16 ± 3,85	92.12 ± 6.42	t = 0.067; *p* = 0.949
**Final Heart Rate (bpm)**	103.38 ± 3.30	118.10 ± 5.81	t = 1.981; *p* = 0.109

Obs. * *p* < 0.05.

**Table 3 ijerph-19-07847-t003:** Results on primary outcomes pre- and post-intervention.

	Experimental Group (*n* = 8)			Control Group (*n* = 6)			MANOVA (2 × 2)
	Pre-Test	Post-Test	% Change	Pre-Test	Post-Test	% Change	
	Mean ± sd	Mean ± sd		Mean ± sd	Mean ± sd		
**PDQ-39 Score Total**	20.66 ± 8.76	21.47 ± 10.50	−3.92	12.05 ± 7.74	10.96 ± 7.33	9.05	F_1,23_ = 0.120; *p* = 0.732
**MDS-UPDRS III**	20.05 ± 13.64	19.55 ± 27.34	2.50	12.36 ± 8.37	7.31 ± 5.16	59.14	F_1,23_ = 2.062; *p* = 0.045 *
**MDS-UPDRS Score Total**	18.53 ± 9.46	9.42 ± 10.40	50.38	10.43 ± 8.36	3.77 ± 2.69	64.95	F_1,23_ = 0.002; *p* = 0.962
**TUG Total Time (s)**	12.53 ± 3.28	11.31 ± 6.06	9.74	10.49 ± 5.63	8.96 ± 3.37	58.59	F_1,23_ = 0.004; *p* = 0.949
**TUG Animals Total Time (s)**	14.22 ± 4.08	17.09 ± 9.93	−20.18	12.32 ± 5.76	10.76 ± 3.87	12.66	F_1,23_ = 0.004; *p* = 0.950
**TUG Numbers Total Time (s)**	14.49 ± 3.03	14.77 ± 6.73	−1.93	12.38 ± 5.37	11.51 ± 3.72	7.03	F_1,23_ = 0.005; *p* = 0.943
**30 Second Chair Stand Test**	11.50 ± 3.25	13.50 ± 3.16	17.39	14.14 ± 2.61	14.67 ± 3.67	3.75	F_1,23_ = 0.691; *p* = 0.415

Obs. PDQ, Parkinson´s disease quality of life questionnaire; MDS-UPDRS, Movement Disorder Society´s Unified Parkinson´s Disease Rating Scale; TUG, Timed Up and Go Test; * *p* < 0.05.

**Table 4 ijerph-19-07847-t004:** Results obtained at the cognitive level.

		Experimental Group 1 (*n* = 8)			Control Group (*n* = 6)			MANOVA (2 × 2)
		Pre-Test	Post-Test	% Change	Pre-Test	Post-Test	% Change	
		Mean ± sd	Mean ± sd		Mean ± sd	Mean ± sd		
**STROOP (*n*)**	**Words**	72.50 ± 12.08	71.63 ± 8.55	1.20	78.43 ± 17.45	81.33 ± 6.65	−3.70	F_1,23_ = 0.021; *p* = 0.886
	**Colors**	44.88 ± 12.05	48.25 ± 13.65	−7.51	62.86 ± 9.35	61.00 ± 5.44	2.96	F_1,23_ = 0.875; *p* = 0.360
	**Colors-Words**	25.50 ± 13.49	29.25 ± 13.64	−14.71	32.00 ± 11.93	31.50 ± 7.69	1.56	F_1,23_ = 0.286; *p* = 0.598
	**Colors-Words Planned**	27.56 ± 6.19	26.95 ± 5.33	2.21	34.48 ± 5.17	34.82 ± 2.79	−0.99	F_1,23_ = 0.008; *p* = 0.929
	**Interference**	−2.06 ± 8.49	2.30 ± 12.08	211.65	−2.48 ± 12.10	−3.32 ± 6.31	−33.87	F_1,23_ = 0.373; *p* = 0.548

Obs. STROOP, Stroop Color-Word Test.

## Data Availability

The data presented in this study are available on request from the corresponding author. The data are not publicly available due to restrictions of privacy or ethical.

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
