# Peer review of "Effects of Cycling Dual-Task on Cognitive and Physical Function in Parkinson’s Disease: A Randomized Double-Blind Pilot Study"

_ijerph, 2022, doi:10.3390/ijerph19137847_

Round 1
Reviewer 1 Report
This manuscript reports the effects of cycling dual-task training on cognitive and physical functions. The dual-task training is an important intervention such as fall prevention, and it is interesting to examine the effect of cycling dual-task on PD patients. However, I have some concerns about the interpretations of results. Please see the specific comments for more detail.
Introduction
1. The motivation of using cycling dual-task is unclear. Why did the authors not use walking, which was commonly used in dual-task studies?
Method
2. Please, describe the method of Stroop task in more detail. How many seconds did the Stroop task run? How did the authors calculate the results of the Stroop task (Interference…etc.)? What is “Colors-Words Planed” (Table 4)?
3. I think that cognitive task group is needed. In the present experimental condition, it is not possible to discuss whether Stroop task performance was improved by the improvement of cognitive function induced by the dual-task training, or cognitive task training included in dual-task training.
Results
4. In the Table 3 and 4, Please provide the definition of “moment 1” and “moment 2”.
5. The author claimed that “a very important change in the EG (211.65%)” (Line 223). However, I think this claim should be judges carefully. First, baseline values for Stroop task differ between groups. Experimental group might have more room for improvement of this task performance. Second, even taking into account the small number of participants, the variability of results in the Stroop task is very large. Is it possible that some participants have a significant impact on the results? Please, present the result of each participant.
Discussion
6. To evaluate mental speed and response inhibition, the authors measured Stroop task performance (in the method section). However, why is this point not mentioned in the discussion section?
7. With this experimental protocol, it is not possible to determine whether the improvement of cognitive task is due to the repetition of the cognitive task, or dual-task intervention.
Author Response
This manuscript reports the effects of cycling dual-task training on cognitive and physical functions. The dual-task training is an important intervention such as fall prevention, and it is interesting to examine the effect of cycling dual-task on PD patients. However, I have some concerns about the interpretations of results. Please see the specific comments for more detail.
 
Introduction
1. The motivation of using cycling dual-task is unclear. Why did the authors not use walking, which was commonly used in dual-task studies?
We have used cycling for several reasons:
- In the first place, we consider that it is safer than treadmill walking for people with Parkinson's disease, since we would have to delimit the sample based on gait.
- The MOTOmed Viva 2 Parkinson is a specific cycle ergometer for patients with PD that allows the evaluation of the patient's stiffness and asymmetries by adapting to the spasms that occur during movement.
- This device maintains a high revolution (90 rpm), which leads to potential improvements in the locomotor system due to the alteration of the functioning of the central nervous system, with a neuroprotective effect.
- This device has been much more comfortable and practical for safe dual tasks in patients with PD.
Ridgel, A. L., Phillips, R. S., Walter, B. L., Discenzo, F. M., & Loparo, K. A. (2015). Dynamic high-cadence cycling improves motor symptoms in Parkinson’s disease. Frontiers in neurology, 6, 194.
Ridgel, A. L., Vitek, J. L., & Alberts, J. L. (2009). Forced, not voluntary, exercise improves motor function in Parkinson’s disease patients. Neurorehabilitation and neural repair, 23 (6), 600-608.
Method
2. Please, describe the method of Stroop task in more detail. How many seconds did the Stroop task run? How did the authors calculate the results of the Stroop task (Interference…etc.)? What is “Colors-Words Planed” (Table 4)?
Thank you so much for pointing out this need for explanation. We have included one more paragraph in the test description with suggested clarifications.
- I think that cognitive task group is needed. In the present experimental condition, it is not possible to discuss whether Stroop task performance was improved by the improvement of cognitive function induced by the dual-task training, or cognitive task training included in dual-task training. 
If we had made 3 groups, the sample size would not be enough to have representative results. In any case, one group performed physical exercise + cognitive tasks, and the control group performed physical exercise. Therefore, we consider that it can be suggested whether or not the dual task has effects on cognition.
Results
4. In the Table 3 and 4, Please provide the definition of “moment 1” and “moment 2”.
It has been changed to pre-test and post-test, and has been reflected in the methodology section to which it refers.
- The author claimed that “a very important change in the EG (211.65%)” (Line 223). However, I think this claim should be judges carefully. First, baseline values for Stroop task differ between groups. Experimental group might have more room for improvement of this task performance. Second, even taking into account the small number of participants, the variability of results in the Stroop task is very large. Is it possible that some participants have a significant impact on the results? Please, present the result of each participant.
The expression has been changed, to be cautious, but the percentage of change is high. As for exposing the result of each participant, we consider that it is not appropriate in this article, because the same would have to be done for each variable. But we can say that all the people who were in the intervention group have greatly improved their results in the Stroop test.
We are aware that by having a non-representative sample of the population, and therefore it is a pilot study, the variability of the results among the participants is important. It is true that when the results of the participants of each group are analyzed individually, a behavior is observed in the results of the test that is not exactly what is observed when considering the results as a group. Even so, we believe it is important to consider them as a group, and observe the results as such, since this study will give rise to a future study with corrections in the sample size so that it is representative of the population and that in this way the possible variability of the subjects within the group may be more diluted, as well as increasing the intervention protocol. We do not consider that the results should be given individually, since this study is characterized as a pilot clinical study and not as a case series study.
Discussion
6. To evaluate mental speed and response inhibition, the authors measured Stroop task performance (in the method section). However, why is this point not mentioned in the discussion section? 
Thank you for your commentary, we have included this explanation in the discussion.
- With this experimental protocol, it is not possible to determine whether the improvement of cognitive task is due to the repetition of the cognitive task, or dual-task intervention.
We agree with the reviewer's opinion that we cannot attribute the improvement of the cognitive component to the cognitive intervention performed in isolation. At no time have we set as the objective of our study to evaluate the effect of a cognitive task in isolation, which may also be interesting in patients with Parkinson's. In our study we intended to evaluate the effect of an isolated physical task on the motor and cognitive components, in addition to assessing the effect of a combined task also on the motor and cognitive components. In summary, how the fact of developing a combined task influences the physical and cognitive results when compared to only a physical task. If there are aspects (physical or cognitive) that can benefit from physical work in isolation or combined work.
Reviewer 2 Report
In this manuscript, the Authors found that Parkinson's disease is associated with Dual-task performance impairment when the subjects perform the cycling and cognitive tasks, with the significant differences in different cycling parameters.
In my opinion, the present paper is interesting and well written. The methodological and statistical procedure is correct. The conclusions are logically linked to the results.
I feel that it is a relevant paper for readers.
I do have some concerns:
1.In the Introduction, as well as in the discussion, the Authors could describe the executive impairment that involves Parkinson's disease, as well as other neurodegenerative diseases (i.e. Alzheimer’s disease, Huntington’s disease), and that can better explain Dual-Task ability impairment.
The Authors could refer to the executive skills than involves, for example, the attentional process, the attentional shifting, and behavioral flexibility.
For example:
Åhman, H. B., Cedervall, Y., Kilander, L., Giedraitis, V., Berglund, L., McKee, K. J., Rosendahl, E., Ingelsson, M., & Åberg, A. C. (2020). Dual-task tests discriminate between dementia, mild cognitive impairment, subjective cognitive impairment, and healthy controls - a cross-sectional cohort study. BMC geriatrics, 20(1), 258. https://doi.org/10.1186/s12877-020-01645-1.
de Oliveira Silva, F., Ferreira, J. V., Plácido, J., & Deslandes, A. C. (2020). Spatial navigation and dual-task performance in patients with Dementia that present partial dependence in instrumental activity of daily living. IBRO reports, 9, 52–57. https://doi.org/10.1016/j.ibror.2020.06.006.
Mancioppi, G., Fiorini, L., Rovini, E., Zeghari, R., Gros, A., Manera, V., Robert, P., & Cavallo, F. (2021). Innovative motor and cognitive dual-task approaches combining upper and lower limbs may improve dementia early detection. Scientific reports, 11(1), 7449. https://doi.org/10.1038/s41598-021-86579-3.
2.The tables can be better formatted. Furthermore, to make the results more readable, the Authors can report only the p-values, eliminating the F and t of the MANOVA and t-test, respectively.
3.Finally, the English language and style are fine, but there are some minor typing or grammar errors. Please carefully check the whole manuscript.
Author Response
In this manuscript, the Authors found that Parkinson's disease is associated with Dual-task performance impairment when the subjects perform the cycling and cognitive tasks, with the significant differences in different cycling parameters.
In my opinion, the present paper is interesting and well written. The methodological and statistical procedure is correct. The conclusions are logically linked to the results.
I feel that it is a relevant paper for readers.
I do have some concerns:
1.In the Introduction, as well as in the discussion, the Authors could describe the executive impairment that involves Parkinson's disease, as well as other neurodegenerative diseases (i.e. Alzheimer’s disease, Huntington’s disease), and that can better explain Dual-Task ability impairment.
The Authors could refer to the executive skills than involves, for example, the attentional process, the attentional shifting, and behavioral flexibility.
For example:
Åhman, H. B., Cedervall, Y., Kilander, L., Giedraitis, V., Berglund, L., McKee, K. J., Rosendahl, E., Ingelsson, M., & Åberg, A. C. (2020). Dual-task tests discriminate between dementia, mild cognitive impairment, subjective cognitive impairment, and healthy controls - a cross-sectional cohort study. BMC geriatrics, 20(1), 258. https://doi.org/10.1186/s12877-020-01645-1.
de Oliveira Silva, F., Ferreira, J. V., Plácido, J., & Deslandes, A. C. (2020). Spatial navigation and dual-task performance in patients with Dementia that present partial dependence in instrumental activity of daily living. IBRO reports, 9, 52–57. https://doi.org/10.1016/j.ibror.2020.06.006.
Mancioppi, G., Fiorini, L., Rovini, E., Zeghari, R., Gros, A., Manera, V., Robert, P., & Cavallo, F. (2021). Innovative motor and cognitive dual-task approaches combining upper and lower limbs may improve dementia early detection. Scientific reports, 11(1), 7449. https://doi.org/10.1038/s41598-021-86579-3.
Thank you very much for your input, we have made changes and expanded the content on executive function impairment in people with Parkinson's disease in the introduction.
2.The tables can be better formatted. Furthermore, to make the results more readable, the Authors can report only the p-values, eliminating the F and t of the MANOVA and t-test, respectively.
Changes have been made to the format of the tables and the content has been adjusted. We consider that the data you ask us to remove is important for the results, which is why we consider not removing it.
3.Finally, the English language and style are fine, but there are some minor typing or grammar errors. Please carefully check the whole manuscript.
Thanks for the comment, we have revised the English to make the relevant changes.
Reviewer 3 Report
Congratulations on the study. You present a study with many variables, with a correct intervention and with a large amount of data that is clearly shown. I suggest including in the methodology if the TUG has been performed with technical assistance and if the patients included in the studies presented it. If not, indicate if it has been an inclusion or exclusion criterion.Author Response
Congratulations on the study. You present a study with many variables, with a correct intervention and with a large amount of data that is clearly shown. I suggest including in the methodology if the TUG has been performed with technical assistance and if the patients included in the studies presented it. If not, indicate if it has been an inclusion or exclusion criterion.
Thanks for the comments about the article.
Patients have performed the TUG without technical assistance, only with the close supervision of a person to avoid a possible fall. At no time has walking with technical assistance been a criterion for inclusion or exclusion of the sample. This criterion has not been assessed as a characteristic of the sample. That is why it has not been named in the article.
Round 2
Reviewer 1 Report
Please, reconsider the following points,1, I think that the results of each participant will help readers interpret the results.
Author Response
Thank you very much for your review and for everything you have contributed to improve the manuscript. Regarding what you ask us in the last revision about showing the results of the participants individually, we consider that it is not a relevant option due to several reasons:
- The sample is too large to show the results individually, which would make it difficult for the reader to understand.
- The results of the participants follow the same trend, so the individualized data will not provide different results.
- The objective of this study is to compare the results between the dual and simple approach, therefore, we are interested in showing it as a comparison between both groups to propose future interventions.
- Finally, we consider that the more data there is, the more difficult it is for the reader to understand the research.
